# Reference genes for mesangial cell and podocyte qPCR gene expression studies under high-glucose and renin-angiotensin-system blocker conditions

**Nicole Dittrich Hosni**[ID]**, Ana Carolina Anauate**[ID]**, Mirian Aparecida Boim**[ID]*

Nephrology Division, Department of Medicine, Escola Paulista de Medicina, Universidade Federal de São Paulo, São Paulo, Brazil

* maboim@unifesp.br

**Data Availability Statement:** All relevant data are within the manuscript.

**Funding:** The present study received funding from Fundação de Amparo à Pesquisa do Estado de São

## Abstract

### Background

Real-time PCR remains currently the gold standard method for gene expression studies. Identification of the best reference gene is a key point in performing high-quality qPCR, providing strong support for results, and performing as a source of bias when inappropriately chosen. Mesangial cells and podocytes, as essential cell lines to study diabetic kidney disease (DKD) physiopathology, demand accurate analysis of the reference genes used thus far to enhance the validity of gene expression studies, especially regarding high glucose (HG) and DKD treatments, with angiotensin II receptor blockers (e.g., losartan) being the most commonly used. This study aimed to evaluate the suitability and define the most stable reference gene for mesangial cell and podocyte studies of an *in vitro* DKD model of disease and its treatment.

### Methods

Five software packages (RefFinder, NormFinder, GeNorm, Bestkeeper, and DataAssist) and the comparative ΔCt method were selected to analyze six different candidate genes: *HPRT*, *ACTB*, *PGAM-1*, *GAPDH*, *PPIA*, and *B2M*. RNA was extracted, and cDNA was synthesized from immortalized mouse mesangial cells and podocytes cultured in 4 groups: control (n = 5; 5 mM glucose), mannitol (n = 5; 30 mM, as osmotic control), HG (n = 5; 30 mM glucose), and HG + losartan (n = 5; 30 mM glucose and $10^{-4}$ mM losartan). Real-time PCR was performed according to MIQE guidelines.

### Results

We identified that the use of 2 genes was the best combination for qPCR normalization for both mesangial cells and podocytes. For mesangial cells, the combination of *HPRT* and *ACTB* presented higher stability values. For podocytes, *HPRT* and *GAPDH* showed the best results.

Paulo (#2015/23345-9 - MAB) and Conselho
Nacional de Desenvolvimento Científico e
Tecnológico (CNPq - NDH). The funders had no
role in study design, data collection and analysis,
decision to publish, or preparation of the
manuscript.

**Competing interests:** The authors have declared
that no competing interests exist.

## Conclusion

This analysis provides support for the use of *HPRT* and *ACTB* as reference genes in mouse
mesangial cell studies of gene expression via real-time PCR, while for podocytes, *HPRT*
and *GAPDH* should be chosen.

## Introduction

Globally, diabetic kidney disease (DKD)-related deaths are increasing compared to other types
of chronic kidney diseases [1]. Diabetes endures as the dominant cause of end-stage renal disease and is responsible for approximately half of cases in developed countries [2].

DKD development triggers glomerular injuries, including hyperfiltration, progressive albuminuria, declining glomerular filtration rate, and eventually end-stage renal disease [3]. Additionally, early cellular damage appears in mesangial cells and podocytes [4]. Characteristic
features of mesangial damage rely on mesangial expansion, cell enlargement, secretion of
extracellular matrix, and ultimately nodular glomerulosclerosis [5]. Commonly, podocytes
exposed to a high glucose environment develop foot process effacement, hypertrophy, detachment from the basal membrane, and apoptosis [6, 7].

Analysis of gene expression in *in vivo* and *in vitro* models of DKD is among the strategies
that contribute to a better understanding of the pathophysiological mechanisms of DKD progression. Quantitative real-time PCR (qPCR) is currently the gold standard method to evaluate
gene expression [8]. Identification of the best reference gene stands as a key point in performing high-quality qPCR, providing strong support for results, as well as acting as a source of
bias when inappropriately chosen. Considering the many steps the procedure goes through
(RNA extraction, reverse-transcription, amplification efficiency, etc.) and the fact that the data
are most frequently relative, not absolute, normalization is established as a critical step to properly standardize the experiment and, thus, provide decisive results for a qPCR assay. Although
the use of reference genes is absolutely acknowledged as the most correct method of normalization, gene choice must be validated according to tissue, cell type, experimental design, and
conditions [9]. There must be a detailed report of the method used to select the most stable
gene and the optimal number of genes recommended [10].

Suitable reference genes have been previously studied for several different models of kidney
disease, such as mouse models of cystic kidney disease, ischemic and toxicological kidney disease in rat models, and kidneys from rats exposed to testosterone [11–13]. Other than mice
and rats, reference genes have been studied in bovine and porcine kidneys [14, 15]. In humans,
165 biopsies from patients with multiple kidney disease diagnostics had their tubule interstitial
compartment microdissected and investigated for the best reference genes in this setting [16].
Regarding glomeruli, a study was performed on microdissected glomeruli of a diabetic rat
model and primary rat mesangial cell culture exposed to high glucose [17]. The latter is the
only available reference in the literature regarding DKD qPCR reference genes. Other specific
cell types from glomeruli do not have support from the literature concerning the best normalization gene for qPCR studies, circumstances that may complicate the interpretation of qPCR
data for researchers in the field, misrepresenting the reliability of the results.

Mesangial cells and podocytes, as essential cell lines in DKD, demand accurate analysis of
the best reference genes to enhance the validity of gene expression studies, especially regarding
high glucose (HG) and different treatments, with angiotensin II receptor blockers being the
most frequently used [18, 19].

Our goal was to evaluate the suitability and define the most stable reference gene specifically for mesangial cell and podocyte studies of an *in vitro* DKD model of disease and its treatment among six commonly used reference genes (*HPRT*, *ACTB*, *PGAM-1*, *GAPDH*, *PPIA*, and *B2M*).

## Materials and methods

### Cell lines and cell culture

Immortalized mesangial cells (SV40 MES 13, ATCC) were cultured in DMEM (Invitrogen Corporation, Gaithersburg, MD, USA) containing 10% fetal bovine serum (FBS), penicillin (50 U/ml) and 2.6 g HEPES at 37˚C. Podocytes (Cell line E11, CLS) were cultured in RPMI 1640 medium (Invitrogen Corporation, Gaithersburg, MD, USA) supplemented with 10% fetal bovine serum and interferon-gamma (*INF*-gama) at 33˚C; after achieving the desired confluence, flasks were transferred to a 37˚C incubator for the differentiation process for 14 days without *INF*-gama. Both cell types were cultured until >90% confluence and remained in a 5% $CO_2$ environment. After 24 hours in 1% FBS, each group received the designated stimulus for 24 hours: pure medium (control group), medium containing 30 mM mannitol (as osmotic control, mannitol group), 30 mM D-glucose (high-glucose group) or 30 mM D-glucose combined with 100 µM losartan (losartan group). The study workflow is shown in Fig 1.

### RNA extraction, quality parameters and reverse transcription

Total RNA was extracted using TRIzol reagent (Life Technologies, USA) according to the manufacturer's instructions. RNA concentration and quality (260/280 ratio >1.8 and 260/230 ratio 2.0–2.2, indicating high purity) were assessed using a NanoVue spectrophotometer (GE Healthcare Life Sciences, USA). RNA integrity was also analyzed by gel electrophoresis. After RNA extraction, we performed DNAse treatment to avoid genomic DNA contamination. Two micrograms of total RNA was reverse-transcribed into cDNA (High Capacity cDNA Reverse Transcription Kit, Applied Biosystems, USA). The reaction mixture was incubated for 10 minutes at 25˚C, 120 minutes at 37˚C and 5 seconds at 85˚C.

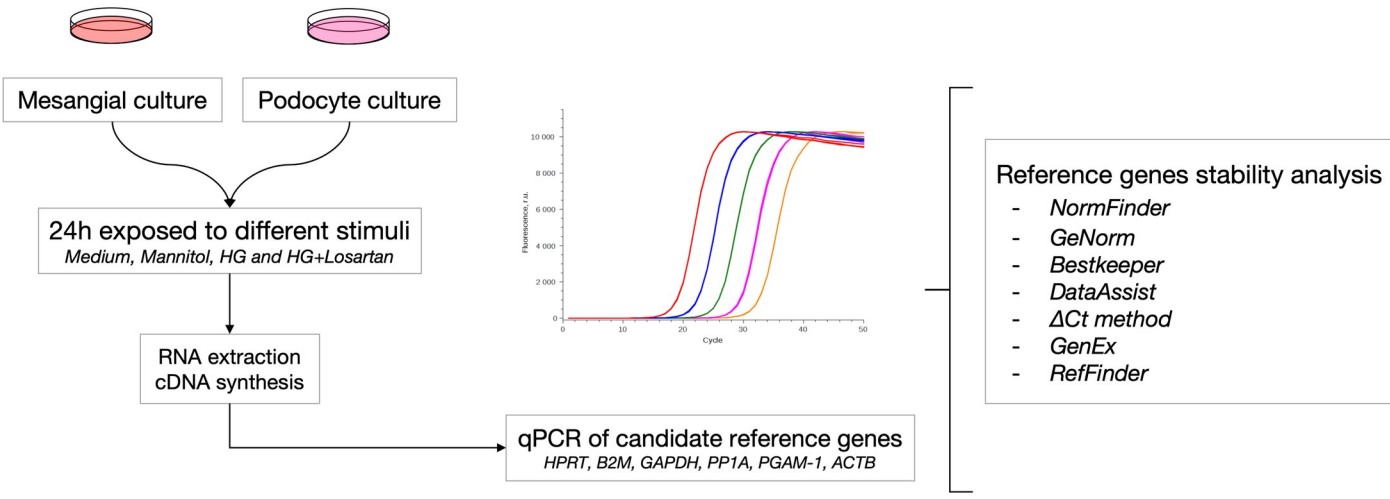

**Fig 1. Study workflow.** The figure shows the workflow for determination of the most stable reference gene for mesangial cells and podocytes exposed to mannitol, high glucose or high glucose and losartan. Five software packages (RefFinder, NormFinder, GeNorm, Bestkeeper, and DataAssist) and the comparative ΔCt method were applied to analyze six different candidate genes: *HPRT*, *ACTB*, *PGAM-1*, *GAPDH*, *PPIA*, and *B2M*. RNA was extracted, and cDNA was synthesized from immortalized mouse mesangial cells and podocytes cultured in 4 groups: control (n = 5; 5 mM glucose), mannitol (n = 5; 30 mM, as osmotic control), HG (n = 5; 30 mM glucose), and HG + losartan (n = 5; 30 mM glucose and $10^{-4}$ mM losartan).

**Table 1. Primer sequences for the six candidate genes.**

| Gene symbol | Target gene | Accession ID | Primer sequence (5'-3') | Amplicon length (bp) |
|---|---|---|---|---|
| ACTB | Beta actin | NM_007393.5 | CGCAGCCACTGTCGAGT | 96 |
| | | | GTCATCCATGGCGAACTGGT | |
| GAPDH | Glyceraldehyde-3-phosphate dehydrogenase | NM_001357943 | GGTGGTCTCCTCTGACTTCAACA | 101 |
| | | | ACCAGGAAATGAGCTTGACAAAG | |
| B2M | Beta-2 microglobulin | NM_009735.3 | ATACGCCTGCAGAGTTAAGC | 70 |
| | | | TCACATGTCTCGATCCCAGT | |
| PPIA | Peptidylprolyl isomerase A | NM_008907.2 | CAGGTCCATCTACGGAGAGA | 146 |
| | | | CATCCAGCCATTCAGTCTTG | |
| HPRT | Hypoxanthine phosphoribosyltransferase | NM_013556.2 | CTCATGGACTGATTATGGACAGGAC | 123 |
| | | | GCAGGTCAGCAAAGAACTTATAGCC | |
| PGAM-1 | Phosphoglycerate mutase 1 | NM_023418.2 | ATCAGCAAGGATCGCAGGTA | 102 |
| | | | TTCATTCCAGAAGGGCAGTG | |

## qPCR performance

Gene expression analysis was performed by qPCR using SYBR Green (Applied Biosystems) in QuantStudio 7 Flex (Applied Biosystems) in accordance with the manufacturer's instructions. Primer sequences for the six genes used are presented in Table 1. The melting curves of all primers are shown in Fig 2. All samples were evaluated in triplicate.

## Software analysis for stability of candidate reference genes

To establish the best reference gene and best combination, we evaluated qPCR results in five different software applications: RefFinder, NormFinder, GeNorm, Bestkeeper, and DataAssist. We also evaluated the data with the comparative ΔCt method.

NormFinder is a freely available tool that provides the stability value for several candidate genes tested on a sample set. Any required number of samples is subject to the analysis, providing an estimation of expression variation [20]. GeNorm software works as an algorithm (M value) to determine the most stable reference genes among a collection of tested candidate

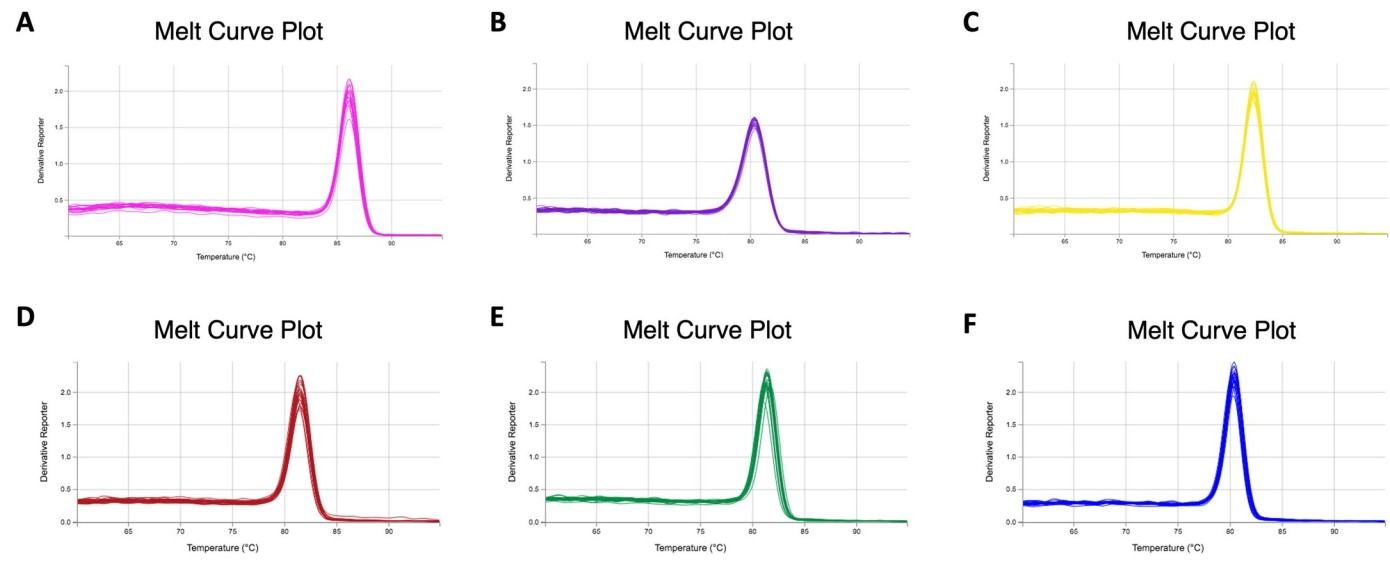

**Fig 2. Melting curves of the primers for the candidate genes.** A) *ACTB*. B) *B2M*. C) *PGAM-1*. D) *GAPDH*. E) *HPRT*. F) *PPIA*.

genes. The tool calculates a normalization factor for each sample, established according to the geometric mean of the reference genes number [10]. Bestkeeper is an Excel-based spreadsheet software that determines the best suited reference genes and combines them into an index, allowing a comparison with further target genes to decide which of them has the best suitability for normalization. The application acknowledges extremely deviating samples that can be removed from the calculation and improves the reliability of the results [21]. DataAssist is an Applied Biosystems software that quantifies relative gene expression across a given number of samples. It provides an "Endogenous Control Selection" tool that shows the Ct values of candidate genes for all samples as well as a score [22]. The ΔCt method compares the relative expression values between 'pairs of genes', implementing an elimination process according to a ranking of the variability among each pair. Subsequently, the most appropriate gene of reference can be selected [23].

The program GenEx was used to calculate the accumulated standard deviation across the samples, providing the necessary number of genes required for the minimum standard deviation [24]. Finally, we used the RefFinder software, an all-encompassing program developed with the aim of evaluating reference genes from experimental data. The tool includes available software algorithms and methods, all of which were previously mentioned: geNorm, Normfinder, BestKeeper, and the comparative ΔCt method. Supported by the ranking of each program, RefFinder calculates the geometric mean for an overall final ranking [25].

## Statistical analysis

The entire dataset was analyzed regarding normality (Shapiro-Wilk test) and homogeneity (Levene's test). All comparisons were analyzed using ANOVA or Kruskal-Wallis tests, according to each test prerequisite. The level of significance considered was $p < 0.05$. Analysis was performed using Jamovi software, version 1.0.1. The results are expressed as the mean ± standard deviation (SD).

## Results

### Expression levels profile of candidate genes of reference

Raw Ct values were acquired in triplicate for both mesangial cell and podocyte samples and analyzed according to each stimulus received. Ct values are inversely proportional to gene expression. The Ct mean of the candidate genes ranged from 29.20 to 18.55 in mesangial cells. The highest Ct among the candidate genes in mesangial cells was achieved by *ACTB* (29.20 ± 1.09), and the lowest was achieved by *PPIA* (18.55 ± 0.79). *HPRT* showed a mean of 23.48 ± 0.97, followed by *GAPDH* (22.46 ± 1.04), *PGAM-1* 21.86 ± 1.06 and *B2M* 18.66 ± 0.78.

For podocytes, otherwise, the mean ranged from 24.45 to 13.02. *ACTB* achieved the highest value (24.45 ± 1.2), while the lowest value was achieved by *B2M* (13.02 ± 0.51). The remaining candidates showed a mean between 19.19 and 14.98: *GAPDH* (19.19 ± 1.00) was followed by *HPRT* (18.85 ± 0.79), *PGAM-1* (17.90 ± 1.16) and *PPIA* (14.98 ± 1.06). The mean Ct value of the triplicates according to each gene and cell line is shown in Fig 3A and 3B.

### Stability of candidate genes

We applied the six algorithms described previously to determine the stability of each reference gene candidate according to the cell type. After analysis with different algorithms and a visual inspection of the ranked genes, we concluded that *HPRT* and *ACTB* for mesangial cells (Table 2) and *HPRT* together with *GAPDH* for podocytes were the best reference genes for qPCR studies (Table 3).

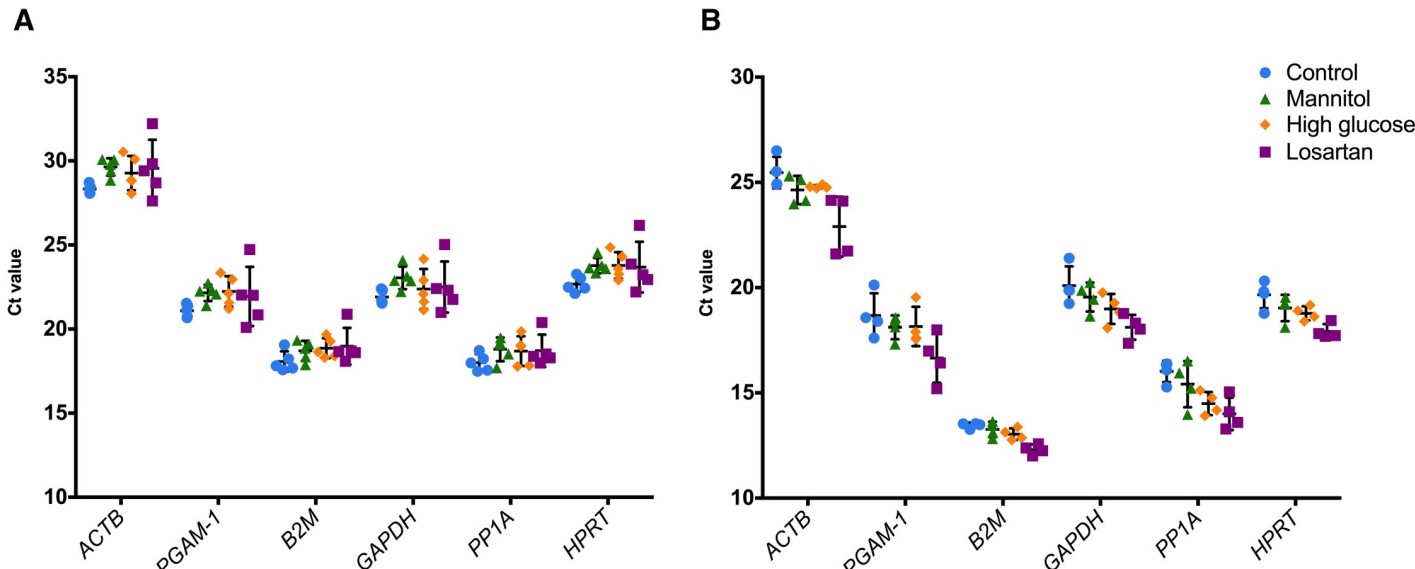

**Fig 3. Expression profile of the six candidate reference genes in mesangial cells (A) and podocytes (B).** A lower threshold value (Ct) represents a higher gene expression level. The data are presented as the mean +/- standard deviation. Each dot represents the average from triplicate ΔCt from each sample. All genes were tested for differences among the groups (Kruskal-Wallis test), and all comparisons showed a nonsignificant result ($p > 0.05$).

NormFinder showed the lowest stability value for *HPRT* for both mesangial cells and podocytes, indicating that this gene is the best reference gene according to this algorithm. As recommended by software instructions, any gene with a stability value higher than 0.5 is considered unstable; all genes tested showed a stability value lower than the cutoff.

Individual results for BestKeeper showed, for mesangial cells, the lowest coefficient of variation (CV) for *ACTB*, indicating that this gene is the best for this cell type under the established conditions according to this software. For podocytes, the lowest CV was displayed for *B2M*, showing up as the most stable option. No candidate gene showed an SD higher than 1.0, the fixed software threshold for instability.

DataAssist software retrieved the PGAM-1 gene as the most stable gene for mesangial cells. Nevertheless, the lowest score for podocytes was achieved by *HPRT*.

**Table 2. Ranking of candidate reference genes by each method used for mesangial cells.**

| NormFinder* | Stability value | GeNorm | M value | BestKeeper | CV [% CP] | std dev [± CP] | DataAssist | Score | RefFinder | Geomean | ΔCt method | Mean SD | Visual inspection** | Frequency |
|---|---|---|---|---|---|---|---|---|---|---|---|---|---|---|
| *HPRT* | 0.118 | *ACTB* | 0.031 | *ACTB* | 2.96 | 0.87 | *PGAM-1* | 5.324 | *HPRT* | 1.00 | *HPRT* | 0.67 | *HPRT* | 3x |
| *ACTB* | 0.158 | *GAPDH* | 0.031 | *HPRT* | 2.99 | 0.70 | *PPIA* | 5.514 | *ACTB* | 2.21 | *ACTB* | 0.75 | *ACTB* | 2x |
| *GAPDH* | 0.179 | *PGAM-1* | 0.034 | *B2M* | 3.03 | 0.57 | *HPRT* | 5.525 | *PPIA* | 3.41 | *PGAM-1* | 0.77 | *PGAM-1* | 1x |
| *PGAM-1* | 0.181 | *HPRT* | 0.038 | *PPIA* | 3.41 | 0.63 | *ACTB* | 5.624 | *PGAM-1* | 3.94 | *PPIA* | 0.77 | *GAPDH* | 1x |
| *PPIA* | 0.226 | *PPIA* | 0.039 | *GAPDH* | 3.51 | 0.79 | *GAPDH* | 5.907 | *GAPDH* | 4.47 | *GAPDH* | 0.86 | *B2M* | 0x |
| *B2M* | 0.244 | *B2M* | 0.042 | *PGAM-1* | 3.65 | 0.80 | *B2M* | 6.835 | *B2M* | 4.56 | *B2M* | 0.92 | *PPIA* | 0x |

Lower values indicate increased stability in gene expression. Each software result is shown in order of stability.

*Best reference genes determined by NormFinder when the intra- and intergroup variations were not considered.

**Visual inspection refers to the number of times each gene appears as the top gene in each analysis.

**Table 3. Ranking of candidate reference genes by each method used for podocytes.**

| NormFinder* | Stability value | GeNorm | M value | BestKeeper | CV [% CP] | std dev [± CP] | DataAssist | Score | RefFinder | Geomean | ΔCt method | Mean SD | Visual inspection** | Frequency |
|---|---|---|---|---|---|---|---|---|---|---|---|---|---|---|
| HPRT | 0.174 | HPRT | 0.030 | B2M | 0.43 | 13.64 | HPRT | 0.54 | HPRT | 1.32 | HPRT | 0.66 | HPRT | 5x |
| GAPDH | 0.204 | GAPDH | 0.030 | HPRT | 0.65 | 20.31 | GAPDH | 0.56 | GAPDH | 2.21 | GAPDH | 0.69 | B2M | 1x |
| PPIA | 0.316 | B2M | 0.037 | GAPDH | 0.79 | 21.40 | PGAM-1 | 0.66 | PGAM-1 | 2.59 | PGAM-1 | 0.79 | GAPDH | 1x |
| PGAM-1 | 0.324 | ACTB | 0.044 | PGAM-1 | 0.82 | 20.13 | PPIA | 0.69 | B2M | 2.83 | B2M | 0.88 | PGAM-1 | 0x |
| B2M | 0.349 | PGAM-1 | 0.049 | ACTB | 0.87 | 26.50 | B2M | 0.72 | PPIA | 5.23 | PPIA | 0.90 | PPIA | 0x |
| ACTB | 0.373 | PPIA | 0.053 | PPIA | 0.88 | 16.51 | ACTB | 0.85 | ACTB | 5.42 | ACTB | 1.01 | ACTB | 0x |

Lower values indicate increased stability in gene expression. Each software result is shown in order of stability.

*Best reference genes determined by NormFinder when the intra- and intergroup variations were not considered.

**Visual inspection refers to the number of times each gene appears as the top gene in each analysis.

Regarding the ΔCt method, the lowest SD was obtained by *HPRT* for both cell lines. The highest SD was shown by *B2M* for mesangial cells and *ACTB* for podocytes, classifying them as the least stable genes according to the method.

For mesangial cells, GeNorm showed the best results for *ACTB* and *GAPDH* together according to M-values. *B2M* was considered the least stable gene. For podocytes, the best pair of M-values was given to *HPRT* and *GAPDH*. The least stable gene for podocytes was *PPIA*.

Based on these results and visual inspection of all data (Tables 2 and 3), *HPRT* was selected as the overall best reference gene for both mesangial cells and podocytes. For mesangial cells, *HPRT* and *ACTB* were considered the best combination of genes for qPCR normalization (Table 2). *PPIA*, otherwise, was classified as the least stable for mesangial cells. Along with *HPRT*, *GAPDH* was also ranked as the most stable candidate reference gene for podocytes, while *ACTB*, *PGAM-1*, and *PPIA* were found to be the least feasible genes (Table 3).

## Determination of the suitable number of reference genes

For each cell line, we determined the optimal number of genes to be used in a gene expression experiment via qPCR. This analysis was performed by Genex software, and the accumulated standard deviation (Acc.SD) parameter was considered for each cell line according to the number of genes used. For mesangial cells, we concluded that the Acc.SD decreased proportionally to the number of genes used. We also observed that the difference from one to two genes was higher than 0.1. However, the difference from two to three genes was smaller than 0.1 –a pattern that could be noticed in the following number of genes as well, achieving a *plateau*. Therefore, it would be reasonable to use two reference genes (*HPRT* and *ACTB*) and maintain a smaller source of error, since a higher number of genes increases the overall noise of the experiment as well as the cost (Fig 4A).

For podocytes, the lowest Acc.SD was acquired in the presence of 2 reference genes (Fig 4B). In this case, the use of 2 genes–*HPRT* and *GAPDH or B2M*, the top genes according to the visual inspection—to analyze qPCR results would be the best option as well. Since *GAPDH* and *B2M* showed the same results on visual inspection ranking, we looked closely to the performance of each gene on all softwares: besides being the top gene for BestKeeper, *B2M* appeared in 5th place for NormFinder, 3rd for GeNorm, 5th for DataAssist, 4th for RefFinder and 4th again for ΔCt method. *GAPDH*, however, in addition to being the top gene in GeNorm, appeared as 2nd for NormFinder, 3rd for BestKeeper, 2nd for DataAssist, 2nd for RefFinder and 2nd for the ΔCt method. Considering the overall performance of both genes, *GAPDH* was selected as the best option to pair with *HPRT* as a reference gene for podocytes.

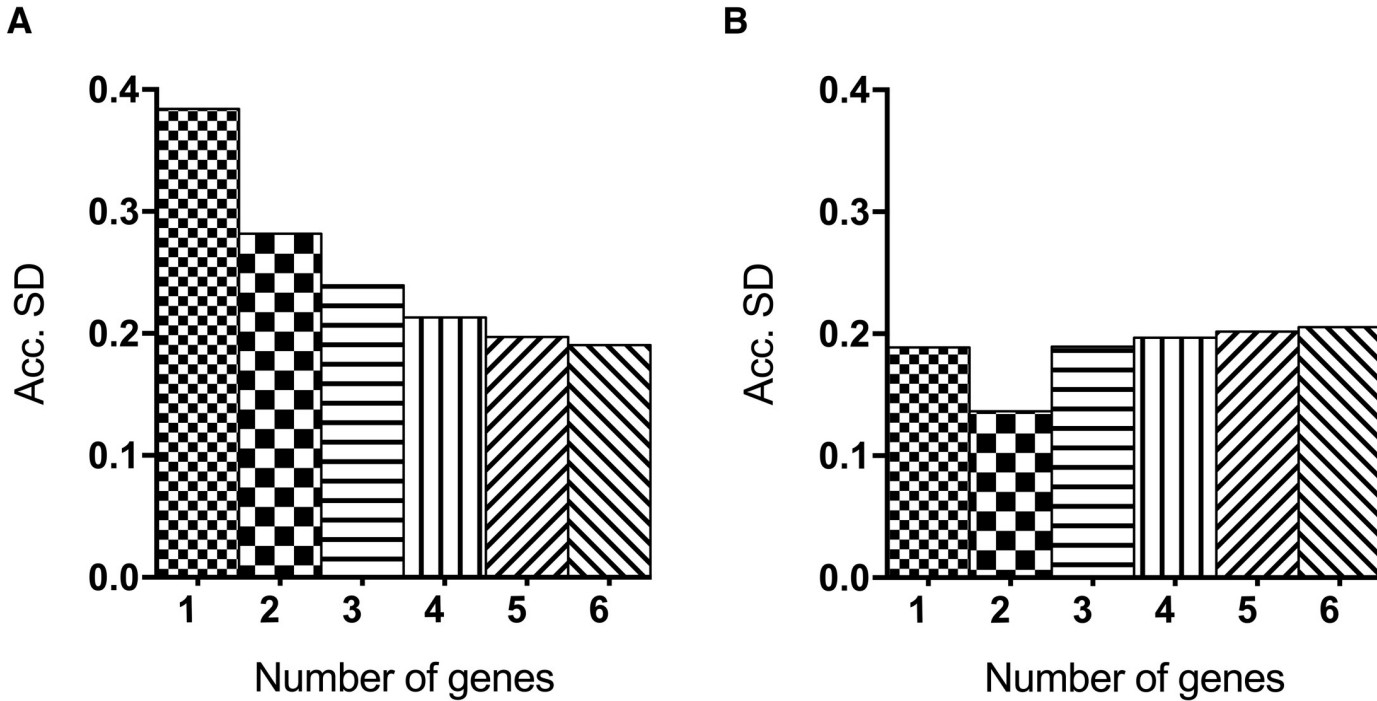

**Fig 4. Evaluation of the optimal number of reference genes in (A) mesangial cells and (B) podocyte cells.** Accumulated standard deviation (Acc.SD) was accessed by GenEx software for the six candidate reference genes in all samples for each cell type. The Acc.SD value was used to determine the optimal number of genes to be used in a gene expression experiment by qPCR. Lowest value of Acc.SD indicate the best number of reference genes.

### Correlation between the top candidates

After determining that the use of 2 reference genes would be the ideal option for mesangial cells and podocytes, we checked if the best 2 genes for each cell line were correlated and therefore could be used simultaneously. We found that the 2 recommended genes for mesangial cells, *HPRT* and *ACTB*, were strongly correlated ($\rho = 0.80$, p<0.0001, Fig 5A), providing support to the recommendation of using these genes at the same time to analyze qPCR data. The same occurred for podocytes: there was a strong correlation between *HPRT* and *GAPDH* expression data, again supporting the use of those genes together as reference genes for qPCR ($\rho = 0.92$, p<0.0001, Fig 5B).

### Validation of the best reference genes

As the results showed that *HPRT* and *ACTB* were the best genes for normalization of mesangial cell qPCR data, we statistically confirmed that there was no difference among the four studied groups regarding the expression of these genes (p>0.05 by Kruskal-Wallis test) (Fig 3A). We also confirmed that there was no difference among the groups of podocytes regarding *HPRT* and *GAPDH* expression, the best genes for this cell type (Fig 3B). In fact, there was no difference between the groups for all candidate genes in either cell line.

### Discussion

The pipeline used in this work has been extensively used throughout many laboratories and is accepted by the literature as a reliable approach to determine the best reference gene to be used, specifically for qPCR in a predetermined biological sample and condition [24, 26, 27]. Here, we aimed to provide data to determine the most suitable reference gene to be used for

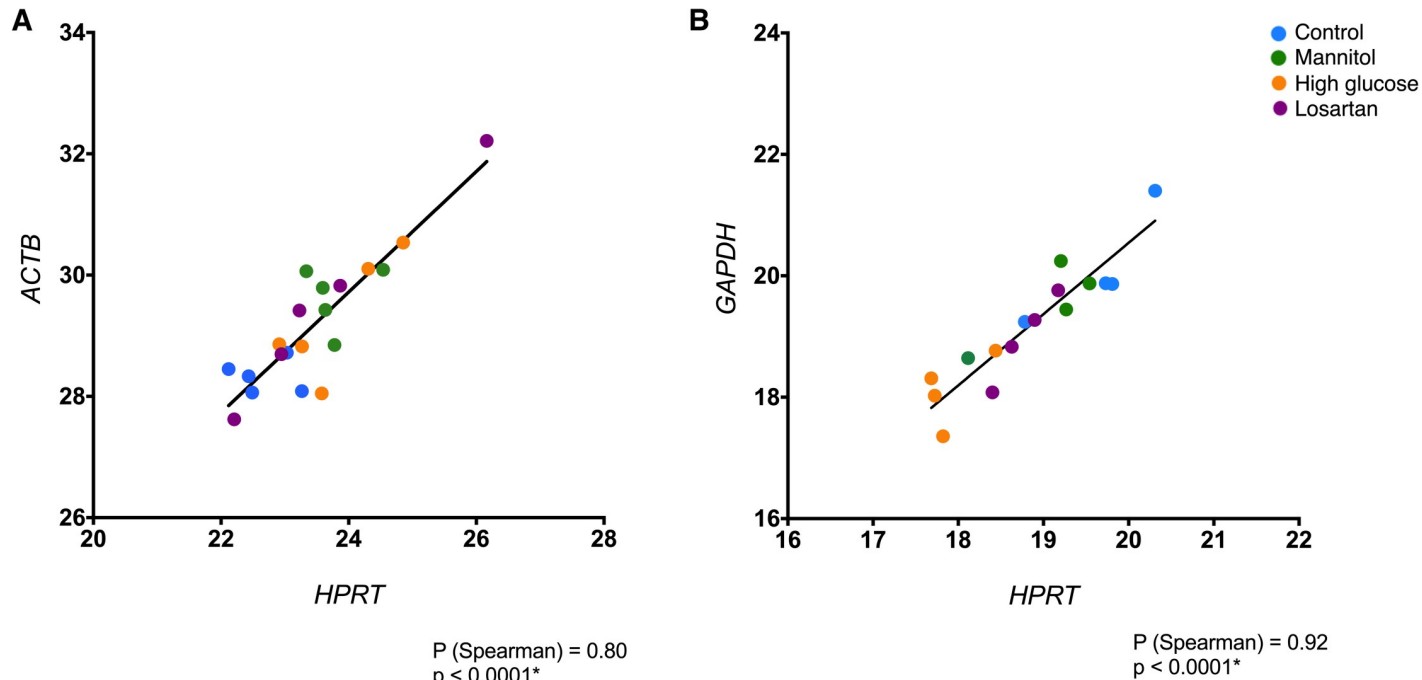

P (Spearman) = 0.80
p < 0.0001*

P (Spearman) = 0.92
p < 0.0001*

**Fig 5. Correlation between top genes for both cell lines.** Highly correlated genes are suitable for simultaneous use as reference genes. A) Correlation between *ACTB* and *HPRT* expression profile in mesangial cells. B) Correlation between *GAPDH* and *HPRT* expression profile in podocytes. ρ: Spearman's rank correlation coefficient. *p<0.0001.

mesangial cells and podocytes exposed to a high glucose environment and treated with losartan, a very known *in vitro* model for diabetic kidney disease [7, 28–31].

Many research groups have clearly shown a need for studies that approach reference genes for their specific study sample [32, 33]. The importance of using the best-known reference gene and pragmatically looking at this question relies on the frequent inappropriate use of the least feasible reference genes, resulting in an inaccurate analysis of qPCR results and therefore in the loss of reagents, time, and samples. Sometimes the most known genes, such as *ACTB* and *GAPDH*, are used for samples and conditions that do not support their use. Even in the most recent years, researchers still normalize their qPCR data of *in vitro* studies based on the most frequently used genes, such as *ACTB* for podocytes and *GAPDH* for mesangial cells (opposite to the finding we had in our analysis), without literature support for this choice [34–39]. Unfortunately, some studies do not clearly provide which reference gene was used to normalize the data, or even if there was data normalization. This shows the need for systematic analysis to identify the best gene or genes to be used as references.

In fact, the literature frequently stands against the use of many popular reference genes. A systematic review performed on vertebrate studies found that 72% of the included studies used *GAPDH*, *ACTB* or *18S* as normalizing genes. The same group shows that as the number of screened reference genes for a specific study design increases, the chance of one of these three genes being the most stable decreases [40].

In nephrology, few studies have addressed reference genes for qPCR normalization [12, 13, 16, 17], exposing a lack of information regarding which gene must be used for gene expression studies for kidney samples and cell lines. The kidney itself is an organ specifically characterized by numerous cell types, justifying the need for reference genes regarding each different cell line [41, 42].

The genes selected as best genes for the studied samples–*HPRT* and *ACTB* for mesangial cells and *HPRT* along with *GAPDH* for podocytes–are extensively described in the literature. Hypoxanthine phosphoribosyltransferase (*HPRT*) is mainly known for its role in the metabolism of purines, although impaired expression of this gene is also responsible for causing cell cycle dysregulation and multisystem regulatory dysfunction [43, 44]. Actin beta (*ACTB*) is involved in cell structure, motility, and integrity, and as it is essential to multiple cell functions, the gene is highly abundant in many cell lines [45]. A previous study on reference genes for rat mesangial cells found *ACTB* to be one of the best reference genes for this cell line under high glucose conditions, in conformity with our findings [17]. Glyceraldehyde 3-phosphate dehydrogenase (*GAPDH*), although it is reported to be involved in cellular survival, apoptosis and DNA repair, is mainly known to express a cellular energy enzyme determinant of the glycolytic process, functioning as a catalyzer of triose phosphate oxidation and, for this reason, ubiquitously distributed in all cell types [46, 47].

The other three genes considered in this study (B2M, PPIA, and PGAM-1) are also known as common reference genes. Beta 2-microglobulin (B2M) expresses a low molecular weight protein related to immune processes linked to the major histocompatibility complex (MHC) [48, 49]. B2M is considered a highly conserved molecule in many different vertebrate species and therefore may be considered a possible reference gene in our setting [49]. Peptidylprolyl isomerase A (PPIA) expresses proteins that catalyze proline imidic peptide bonds in oligopeptides and are also implicated in protein folding processes [50]. It is ubiquitously distributed in multiple cell types, including kidney cells [51]. Phosphoglycerate mutase 1 (PGAM-1) expresses an enzyme responsible for catalyzing 3-phosphoglycerate (3-PGA) to 2-phosphoglycerate (2-PGA), playing an essential role in glycolysis. It is also ubiquitously distributed in multiple cell types [52, 53]. Shared characteristics related to vital processes in the organism, such as glycolysis, immune response, protein folding and cell structure, especially due to applicability to multiple species and most tissues, bring up those 6 genes as relevant targets to investigate as suitable reference genes in our study.

As long-established cell lines in the literature, mesangial cells and podocytes are important biological samples to determine the best reference gene–many researchers in the field are focused on these structures [54–59], and the data provided by our work could potentially influence many studies, providing support to avoid incorrect interpretation of results and their influence in downstream analysis and further conclusions.

## Conclusion

We analyzed six different genes using five software applications and the ΔCt method to determine that the best genes to be used for mesangial cell studies with high glucose and angiotensin receptor II blockers are *HPRT* and *ACTB*, while under the same conditions, the best combination of genes for podocyte gene expression normalization is *HPRT* together with *GAPDH*. We believe our work may provide support to many research laboratories engaged in mesangial cell and podocyte cell culture studies, allowing them to improve the quality of gene expression studies via qPCR and, consequently, the overall quality of nephrology research.

## Acknowledgments

We would like to thank Antonio S. Novaes for cell culture and qPCR technique training.

## Author Contributions

**Conceptualization:** Nicole Dittrich Hosni, Ana Carolina Anauate, Mirian Aparecida Boim.

**Data curation:** Nicole Dittrich Hosni, Ana Carolina Anauate, Mirian Aparecida Boim.

**Formal analysis:** Nicole Dittrich Hosni.

**Funding acquisition:** Mirian Aparecida Boim.

**Investigation:** Nicole Dittrich Hosni.

**Methodology:** Nicole Dittrich Hosni, Ana Carolina Anauate.

**Project administration:** Nicole Dittrich Hosni, Mirian Aparecida Boim.

**Resources:** Mirian Aparecida Boim.

**Software:** Nicole Dittrich Hosni, Ana Carolina Anauate.

**Supervision:** Mirian Aparecida Boim.

**Validation:** Nicole Dittrich Hosni.

**Visualization:** Nicole Dittrich Hosni.

**Writing – original draft:** Nicole Dittrich Hosni.

**Writing – review & editing:** Nicole Dittrich Hosni, Ana Carolina Anauate, Mirian Aparecida Boim.

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
