## [Decision Letter · Decision Letter 0]

19 Apr 2021

PONE-D-21-01454

Reference genes for mesangial cell and podocyte qPCR gene expression studies under high-glucose and renin-angiotensin-system blocker conditions

PLOS ONE

Dear Dr. Boim,

Thank you for submitting your manuscript to PLOS ONE. After careful consideration, we feel that it has merit but does not fully meet PLOS ONE’s publication criteria as it currently stands. Therefore, we invite you to submit a revised version of the manuscript that addresses the points raised during the review process.

We look forward to receiving your revised manuscript.

Kind regards,

Muhammad Shareef Masoud, Ph.D

Academic Editor

PLOS ONE

Journal Requirements:

Reviewers' comments:

Reviewer's Responses to Questions

**Comments to the Author**

1. Is the manuscript technically sound, and do the data support the conclusions?

Reviewer #1: Partly

Reviewer #2: Yes

2. Has the statistical analysis been performed appropriately and rigorously? 

Reviewer #1: Yes

Reviewer #2: Yes

3. Have the authors made all data underlying the findings in their manuscript fully available?

Reviewer #1: No

Reviewer #2: Yes

4. Is the manuscript presented in an intelligible fashion and written in standard English?

Reviewer #1: No

Reviewer #2: No

5. Review Comments to the Author

Reviewer #1: 1: A certain view is implied by the authors regarding to this study but they could be more explicit.

The authors take a rather narrow view of data publication, which I think hinders their analyses. In this study, author need to elaborate further manuscript methodology and results and results should be reconfirmed with other assay as well.

2: Manuscript writeup should improve.

3: Figure legends could not show proper description, So its better to write in detail.

Reviewer #2: In this manuscript entitled Reference genes for mesangial cell and podocyte qPCR gene expression studies under high-glucose and renin-angiotensin-system blocker conditions analyze six genes (HPRT, ACTB, PGAM-1, GAPDH, PPIA, and B2M) using five software applications and the ΔCt method to determine that the best genes to be used for mesangial cell studies with high glucose and angiotensin receptor II blocker are HPRT and ACTB, while in the same conditions, the best combination of genes for podocyte gene expression normalization is HPRT. The study described in the manuscript is interesting however the manuscript can be considered for publication only if the following issues are addressed:

Major Comments

1 Why authors select these six genes (HPRT, ACTB, PGAM-1, GAPDH, PPIA, and B2M). Please elaborate in this article with references

2 Authors should also compare the results of syber green with probe based real time PCR

3. Grammatical errors in all parts of manuscript

6. PLOS authors have the option to publish the peer review history of their article (what does this mean?). If published, this will include your full peer review and any attached files.

Reviewer #1: No

Reviewer #2: No

---

## [Author Response · Author response to Decision Letter 0]

1 Jun 2021

Reviewer #1

• A certain view is implied by the authors regarding to this study but they could be more explicit. The authors take a rather narrow view of data publication, which I think hinders their analyses. 

o We would like to thank this Reviewer for the time dedicated to evaluate this manuscript. We have reviewed the introduction and discussion sections to strengthen the background and rationale of our study, especially including previous data published from other groups.

• Results should be reconfirmed with other assay as well.

o We appreciate the Reviewer’s comment. Literature has shown that probe-based PCR can be adequately reproduced by SYBR Green real-time PCR, delivering highly comparable results with TaqMan and other quantitative gene expression methods (Tajadini 2014, 10.4103/2277-9175.127998; Arikawa 2008, 10.1186/1471-2164-9-328). Other than that, our melting curve analysis showed neither unspecific products nor primer dimers. We also made sure to use proper negative controls for all reactions performed. Multiple studies specifically on reference genes have been performed based on SYBR Green assays only (Zhang 2017, 10.3892/ol.2017.7002; Adeola 2018, 10.4314/ejhs.v28i6.9; Hashemi 2021, 10.1016/j.bbrc.2012.09.009; Nygard 2007, 10.1186/1471-2199-8-67; Ahn 2008, 10.1186/1471-2199-9-78; Bokhale 2020, 10.1016/j.dib.2020.105750). Considering all this information, we believe our results are reliable.

• Manuscript writeup should improve. 

o We appreciate the Reviewer's suggestion. We submitted the manuscript to standard English grammatical revision before resubmission of this revised manuscript to PLOS ONE.

• Figure legends could not show proper description, So its better to write in detail.

o We agree with the Reviewer’s comment, and we have altered the figure legends by including details, as suggested.

Reviewer #2

• In this manuscript entitled Reference genes for mesangial cell and podocyte qPCR gene expression studies under high-glucose and renin-angiotensin-system blocker conditions analyze six genes (HPRT, ACTB, PGAM-1, GAPDH, PPIA, and B2M) using five software applications and the ΔCt method to determine that the best genes to be used for mesangial cell studies with high glucose and angiotensin receptor II blocker are HPRT and ACTB, while in the same conditions, the best combination of genes for podocyte gene expression normalization is HPRT. The study described in the manuscript is interesting however the manuscript can be considered for publication only if the following issues are addressed.

o We would like to thank this Reviewer's careful reading and attention to our work. We have addressed the issues as the following:

• Why authors select these six genes (HPRT, ACTB, PGAM-1, GAPDH, PPIA, and B2M). Please elaborate in this article with references.

o We agree with the Reviewer’s comment. We have included the rationale of each gene included in our study in the Discussion section, with references.

• Authors should also compare the results of syber green with probe based real time PCR

o We appreciate the Reviewer’s comment. Literature has shown that probe-based PCR can be adequately reproduced by SYBR Green real-time PCR, delivering highly comparable results with TaqMan and other quantitative gene expression methods (Tajadini 2014, 10.4103/2277-9175.127998; Arikawa 2008, 10.1186/1471-2164-9-328). Other than that, our melting curve analysis showed neither unspecific products nor primer dimers. We also made sure to use proper negative controls for all reactions performed. Multiple studies specifically on reference genes have been performed based on SYBR Green assays only (Zhang 2017, 10.3892/ol.2017.7002; Adeola 2018, 10.4314/ejhs.v28i6.9; Hashemi 2021, 10.1016/j.bbrc.2012.09.009; Nygard 2007, 10.1186/1471-2199-8-67; Ahn 2008, 10.1186/1471-2199-9-78; Bokhale 2020, 10.1016/j.dib.2020.105750). Considering all this information, we believe our results are reliable.

• Grammatical errors in all parts of manuscript.

o We appreciate the Reviewer's suggestion. We submitted the manuscript to standard English grammatical revision before resubmission of this revised manuscript to PLOS ONE.

---

## [Editor Report · Decision Letter 1]

22 Jun 2021

Reference genes for mesangial cell and podocyte qPCR gene expression studies under high-glucose and renin-angiotensin-system blocker conditions

PONE-D-21-01454R1

Dear Dr. Boim,

We’re pleased to inform you that your manuscript has been judged scientifically suitable for publication and will be formally accepted for publication once it meets all outstanding technical requirements.

Kind regards,

Muhammad Shareef Masoud, Ph.D

Academic Editor

PLOS ONE

---

## [Editor Report · Acceptance letter]

28 Jun 2021

PONE-D-21-01454R1 

Reference genes for mesangial cell and podocyte qPCR gene expression studies under high-glucose and renin-angiotensin-system blocker conditions 

Dear Dr. Boim:

I'm pleased to inform you that your manuscript has been deemed suitable for publication in PLOS ONE. Congratulations! Your manuscript is now with our production department. 

Kind regards, 

on behalf of

Dr. Muhammad Shareef Masoud 

Academic Editor

PLOS ONE